# Generation of Highly Efficient Equine-Derived Antibodies for Post-Exposure Treatment of Ricin Intoxications by Vaccination with Monomerized Ricin

**DOI:** 10.3390/toxins10110466

**Published:** 2018-11-12

**Authors:** Reut Falach, Anita Sapoznikov, Ron Alcalay, Moshe Aftalion, Sharon Ehrlich, Arik Makovitzki, Avi Agami, Avishai Mimran, Amir Rosner, Tamar Sabo, Chanoch Kronman, Yoav Gal

**Affiliations:** 1Department of Biochemistry and Molecular Genetics, Israel Institute for Biological Research, Ness-Ziona 76100, Israel; reutf@iibr.gov.il (R.F.); anitas@iibr.gov.il (A.S.); rona@iibr.gov.il (R.A.); moshea@iibr.gov.il (M.A.); sharone@iibr.gov.il (S.E.); tamars@iibr.gov.il (T.S.); chanochk@iibr.gov.il (C.K.); 2Department of Biotechnology, Israel Institute for Biological Research, Ness-Ziona 76100, Israel; arikm@iibr.gov.il (A.M.); avia@iibr.gov.il (A.A.); avishaim@iibr.gov.il (A.M.); 3Veterinary Center for Preclinical Research, Israel Institute for Biological Research, Ness-Ziona 76100, Israel; amirr@iibr.gov.il

**Keywords:** ricin, vaccine, antitoxin, RTA, RTB, reduction, alkylation

## Abstract

Ricin, a highly lethal toxin derived from the seeds of *Ricinus communis* (castor beans) is considered a potential biological threat agent due to its high availability, ease of production, and to the lack of any approved medical countermeasure against ricin exposures. To date, the use of neutralizing antibodies is the most promising post-exposure treatment for ricin intoxication. The aim of this work was to generate anti-ricin antitoxin that confers high level post-exposure protection against ricin challenge. Due to safety issues regarding the usage of ricin holotoxin as an antigen, we generated an inactivated toxin that would reduce health risks for both the immunizer and the immunized animal. To this end, a monomerized ricin antigen was constructed by reducing highly purified ricin to its monomeric constituents. Preliminary immunizing experiments in rabbits indicated that this monomerized antigen is as effective as the native toxin in terms of neutralizing antibody elicitation and protection of mice against lethal ricin challenges. Characterization of the monomerized antigen demonstrated that the irreversibly detached A and B subunits retain catalytic and lectin activity, respectively, implying that the monomerization process did not significantly affect their overall structure. Toxicity studies revealed that the monomerized ricin displayed a 250-fold decreased activity in a cell culture-based functionality test, while clinical signs were undetectable in mice injected with this antigen. Immunization of a horse with the monomerized toxin was highly effective in elicitation of high titers of neutralizing antibodies. Due to the increased potential of IgG-derived adverse events, anti-ricin F(ab’)_2_ antitoxin was produced. The F(ab’)_2_-based antitoxin conferred high protection to intranasally ricin-intoxicated mice; ~60% and ~34% survival, when administered 24 and 48 h post exposure to a lethal dose, respectively. In line with the enhanced protection, anti-inflammatory and anti-edematous effects were measured in the antitoxin treated mice, in comparison to mice that were intoxicated but not treated. Accordingly, this anti-ricin preparation is an excellent candidate for post exposure treatment of ricin intoxications.

## 1. Introduction

Ricin, a type II ribosome inactivating protein (RIP) derived from the seeds of *Ricinus communis* (castor beans), consists of two polypeptide subunits (A and B) linked by a disulfide bond. The B subunit (RTB) is a lectin, which binds to galactose residues on the cell surface, enabling toxin internalization into the cells. The catalytically active A subunit (RTA) is translocated into the cytoplasm, where it depurinates a conserved adenine residue within the 28S ribosomal RNA of the 60S subunit, leading to irreversible inhibition of protein synthesis and ultimately to cell death [1]. Classified as a Category B agent by the U.S. Centers for Disease Control and Prevention (CDC), ricin is considered a potential bioterror agent mainly due to its high availability and ease of preparation [2]. Ricin toxicity depends on the route of exposure, inhalational and parenteral being highly fatal. Although prophylactic anti-ricin vaccines are being developed [3], the only post-exposure measure found effective against ricin intoxications in pre-clinical settings, is passive immunization with anti-ricin neutralizing antibodies [4,5,6,7]. Anti-ricin antibodies may be elicited following vaccination of various animal species, including mice [8], rabbits [9], monkeys [10], horses [11] and sheep [12]. A variety of ricin immunogens were employed to elicit neutralizing antibody responses against ricin. A toxoid-based vaccine (formaldehyde-inactivated ricin) was shown to induce high titers of protective antibodies both in rabbits [4] and in sheep [12]. Anti-ricin preparations were reported to elicit potent toxin neutralization in vitro and in vivo following horse immunization with an RTA/RTB chain construct, in which the native inter-chain linking domain has been replaced by a non-cleavable linker [11]. Vaccination of animals using the native ricin toxin emulsified in adjuvant was also effective in the elicitation of high titers of neutralizing antibodies [13].

As part of ongoing efforts to develop novel yet safe vaccination strategies that will induce high titers of ricin neutralizing antibodies, we established a method for ricin subunit-based immunization following irreversible monomerization of the toxin to its RTA and RTB constituents. The antigen was prepared by treating the toxin with a reducing agent to sever the inter-subunit covalent bond, and then alkylating the monomeric toxin subunits to prevent their re-dimerization, thereby generating a stable monomerized ricin preparation for animal immunization with substantially reduced toxicity.

In the present study, the efficacy potential of the monomerized ricin vaccine was demonstrated in rabbits, after which the antigen, which was produced at large amount, was thoroughly characterized. The antigen, which was used for vaccinating the horse, was not only safe, but also elicited high titers of highly potent neutralizing antibodies against ricin. Passive immunization with the F(ab’)_2_-based antitoxin conferred high protection against a lethal intranasal ricin challenge at clinically relevant treatment time points following intoxication, and displayed significant anti-inflammatory and anti-edematous effects.

## 2. Results

### 2.1. Elicitation of Anti-Ricin Antibodies Following Rabbit Immunization with Monomerized Ricin

To determine whether the deactivation of ricin by monomerization, affects its ability to elicit neutralizing antibodies, we compared the anti-ricin antibody titers elicited by immunization either with the monomeric antigen or with native holotoxin. To this end, rabbits were immunized at 4-week intervals, with increasing antigen doses reaching up to 100 µg antigen/rabbit, and hyperimmune sera collected at 16 weeks after the first immunization dose were characterized. As seen (Table 1) high ELISA- and neutralizing-antibody titers were reached following immunization with both native or monomerized ricin (2.56 × 10^5^ vs. 4.8 × 10^5^ ELISA units; 7.68 × 10^4^ vs. 9.6 × 10^4^ neutralizing units, respectively). In line with these findings, hyperimmune sera collected from rabbits immunized with either monomerized or native ricin, conferred similarly high level protection in vivo; when mice were treated intramuscularly (i.m.) with the rabbit antisera 24 h prior to i.m. ricin challenge, the doses which conferred 50% protection (PD_50_) were 0.8 and 1.1 µL/mouse, respectively. These experiments suggest that immunization of a horse with monomerized ricin vaccine would be as effective as immunization with native toxin in elicitation of neutralizing anti-ricin antibodies, while the neutralized monomer-based antigen would possess much greater safety margins.

### 2.2. Characterization of the Monomerized Ricin Vaccine

#### 2.2.1. Analytical Assessment

As the monomerized ricin antigen proved to elicit high anti-ricin antibody titers in rabbits, which conferred high level protection to ricin-intoxicated mice, we prepared deactivated monomerized ricin antigen at large amount for the immunization of a horse while carefully monitoring the various steps comprising the ricin purification and monomerization process (Figure 1A). The source material, crude ricin, displays a two-main band appearance of approximately 120 and 65 kDa on SDS-PAGE, representing *Ricinus communis* agglutinin (RCA) and ricin, respectively (lane 2), while the gel-filtration purified ricin appears as a single ~65 kDa band (lane 3) and the monomerized ricin subunits, generated by reduction and alkylation of purified ricin, appear as a ~30 kDa band doublet, representing RTA and RTB (lane 4). Next, we examined whether the purified subunits retain the RTA-dependent catalytic and RTB-dependent lectin activities. To this end, the monomerized subunit mixture was applied to an α-Lactose-Agarose column. The monomerized ricin Ultra Performance Liquid Chromatography (UPLC) chromatogram (Figure 1(B-1)), contains three main peaks, at 150, 170 and 210 s. The flow through contained only the last major peak (Figure 1(B-2)), which could be related to alkylated-RTA. The two other peaks that were attached to the column were collected only upon elution with 0.5 M galactose (Figure 1(B-3)), indicating that these two peaks represent the RTB subunit, which retains its lectin activity. To appreciate the functionality of alkylated-RTA, we assessed the activity of the alkylated-toxin in a cell free system using the reticulocyte lysate-based transcription and translation (TnT) assay. In the TnT assay (Figure 1C), wherein the toxin’s activity is quantified by its ability to inhibit ribosomal translation of mRNA encoding for luciferase enzyme [14,15], not only did monomerized ricin prevent luciferase from being synthesized (ED_50_ = 0.6 ng/mL), it was also found to be a more potent inhibitor (~7 fold) than native ricin (ED_50_ = 3.9 ng/mL). Preservation of the biological activity of the monomerized ricin subunits would imply that their structural conformations did not alter in any significant manner during the monomerization process. The functional conservation of the biological activities of the two alkylated subunits may lead one to expect that antibodies raised against the toxin monomers, will interact with the holotoxin, a mandatory prerequisite for passive immunization against ricin intoxications.

#### 2.2.2. In-Vitro and In-Vivo Toxicity of the Monomerized Ricin

Unlike the catalytic activity of ricin in acellular expression systems such as the TnT-based analysis described above, which is dependent on the RTA unit itself, the catalytic activity of ricin in a cell-based system, requires the presence of intact dimeric toxin molecules, since it is the lectin function of the attached RTB which enables RTA internalization into cells. To evaluate the residual cellular toxicity of the monomerized ricin, cultured HEK-293-acetylcholinesterase (AChE) cells, which produce and secrete recombinant AChE to the medium in a constitutive manner [16], were incubated with different concentrations of native or monomerized ricin, and secreted AChE levels were determined. As seen clearly (Figure 1D), protein (AChE) synthesis inhibition by monomerized ricin was dramatically reduced, the IC_50_ of monomerized ricin being 260 fold higher than that of native ricin (4729 ± 573 versus 18 ± 3 pg/mL, respectively), indicating a loss-of-activity of >99.5%. Following this in vitro assay and prior to horse immunization, we conducted an in vivo safety test in mice. To this end, 40 µg/kg monomerized-ricin (a 10-fold higher dose than the intended initial equine-vaccination dose) was injected intraperitoneally to mice, and body weights were monitored for 14 days. No body weight loss was recorded within this period of time (Figure 1E), nor were any noticeable side effects observed (data not shown).

### 2.3. Anti-Ricin Titer Buildup Following Horse Immunization

Although the antigen was shown to be safe, as an extra precaution, we devised a vaccination protocol for immunizing a horse, comprising a low initial dose, followed by increasing doses of the monomerized ricin antigen. Serum samples were collected three weeks after each injection, to determine ELISA and neutralizing Ab titer buildup (Figure 2). Repeated vaccinations at intervals of 3 weeks resulted in increasing antibody titers up to 11 weeks, after which antibody titers began to level off. To induce a robust immunological response, we discontinued boosting while monitoring antibody titers on a monthly basis. During a period of 3 months in which the horse was not immunized, titer levels significantly decreased, from ~10^5^ to ~10^4^, after which immunization was recommenced with monthly doses of 10 mg monomerized ricin. This immunization regime led to the generation of anti-ricin antibodies at titer levels that were almost one order of magnitude higher than the titer before resuming immunization. Thus, following a vaccination period of 25 weeks, high and stable ELISA- and neutralizing-antibody titers of 0.64 × 10^6^ and 1.28 × 10^6^ units/mL, respectively, were reached.

### 2.4. In Vitro and In Vivo Efficacy of the Anti-Ricin Antitoxin

Pooled horse hyperimmune plasma served as the source material for the production of concentrated anti-ricin F(ab’)_2_-based antitoxin. The motivation to use a F(ab’)_2_ fragment as an antitoxin, was the potential of equine IgG antibodies to cause serum sickness [17], as the crystallizable fragment (Fc) present in IgG, may induce inflammation and unwarranted immune responses [18,19]. To characterize the F(ab’)_2_-based antitoxin, we evaluated its neutralizing potency in the cultured HEK-293-AChE cell system. To this end, ricin was mixed with increasing concentrations of anti-ricin F(ab’)_2_ and then added to the cells. As seen (Figure 3A), the antitoxin inhibited the activity of ricin and restored protein synthesis in a dose dependent manner. The antitoxin dose needed to neutralize 50% (ED_50_) of the toxin was approximately ~0.7 nM. 

To determine the efficacy of this equine-derived antitoxin, mice were intranasally challenged with a lethal dose of ricin, and 24 h later were subjected to F(ab’)_2_ antitoxin treatment via the intranasal route. Overall (Figure 3B), the treatment led to significantly high surviving ratios (62%). Since the clinical treatment of ricin-intoxicated subjects is expected to be via the intravenous route, an additional group was treated intravenously. As can be seen, the surviving ratios (65%) are practically the same as those obtained following intranasal treatment. To determine the efficacy of a late time intervention, mice were intranasally intoxicated with ricin and treated intravenously 48 h post exposure with the same horse antitoxin. Even at this late time point, considerable surviving ratios (34%) were obtained.

### 2.5. Horse Antitoxin Attenuates Ricin-Induced Pulmonary Damage Markers 

To determine the effect of horse derived F(ab’)_2_ anti-ricin antitoxin administration on pathological markers, intranasally-intoxicated mice (7 µg ricin/kg) were intravenously treated 24 h later with the antitoxin and bronchoalveolar lavage fluids (BALFs) collected at 72 h post-intoxication were analyzed for damage markers. A prominent hallmark of pulmonary ricinosis in mice is the presence of exceptionally high levels of the pro-inflammatory cytokine interleukin-6 (IL-6) in BALF [20,21,22,23]. Remarkably (Table 2), treatment with the horse derived F(ab’)_2_ anti-ricin antitoxin induced a sharp attenuation in IL-6 levels (~90% reduction); ~450 and ~3550 pg/mL of IL-6 were measured in the BALFs of antitoxin-treated and non-treated ricin-intoxicated mice, respectively.

Altered lung fluid balance, leading to increased permeability pulmonary edema, is a major pathophysiological characteristic of intranasal ricin intoxication and high levels of protein were reported to be present in the BALF sampled from the inflamed lungs [6,21,22], indicating that the lung–blood barrier has been disrupted. Accordingly, edema markers in mice lungs were assessed. In addition to overall protein level, we have previously demonstrated that increased pulmonary edema can be well monitored by determining cholinesterase (ChE) levels in BALF. Normally, ChE is confined to the bloodstream, yet appears at elevated levels in the BALF following disruption of the pulmonary epithelial–endothelial barrier [20]. In the antitoxin-treated mice, total protein and ChE measurements in BALF collected at 72 h post-exposure, were significantly lower in comparison to untreated mice (63% reduction in total protein, from ~5.5 to ~2 mg/mL, and 78% reduction in ChE, from ~300 to ~70 mU/mL).

Previous studies carried out in our laboratory established that xanthine oxidase (XO), an oxidative stress marker, which may also contribute to edema formation, is dramatically elevated in BALFs of mice intranasally intoxicated with ricin, at 72 h post exposure [20,21,22]. We therefore measured XO levels in BALFs of ricin intoxicated mice that were treated with the horse antitoxin. Indeed, XO levels were significantly reduced, by ~60% following antitoxin administration (~1.7 and ~4.0 mU/mL were measured in BALFs of antitoxin-treated and non-treated ricin-intoxicated mice, respectively) (Table 2).

## 3. Discussion

In the present study, a neutralized monomer-based ricin vaccine was generated by reduction and alkylation of the disulfide bond linking RTA to RTB, in order to immunize a horse for the production of highly potent neutralizing anti-ricin antibodies.

After demonstrating in rabbits that immunization with monomerized ricin was as effective as immunization with native ricin in elicitation of anti-ricin neutralizing antibodies, a thorough characterization of the antigen was conducted. Previous studies demonstrated that reduction of disulfide bonds might abolish the capability to produce biologically active antibodies, as in the case of *Plasmodium falciparum* merozoite surface glycoprotein (gp195). Reduction and alkylation of gp195 triggered an inappropriate folding of the unbound subunits, resulting in a drastic conformational change and significantly altered antigenicity [24]. Thus it was essential to demonstrate that monomerized ricin, which would be repeatedly administered to the horse over a considerably long period of time, retains its structural activity thereby implying proper folding and potential antigenicity of the irreversibly-separated subunits. Indeed, the monomeric alkylated-RTB bound firmly to an α-lactose agarose column, and eluted only following 0.5 M galactose addition, confirming RTB lectin activity. The catalytic activity of the alkylated-RTA, namely protein synthesis arrest in the TnT assay, was not only retained but even was superior to that of native RTA. This is probably due to a limited, or non-complete, separation of the holotoxin subunits of native ricin following reduction, and the relative proximity between the reduced native subunits leading to sporadic reconstitution, in contrast to the complete, irreversible, separation of the monomerized-ricin subunits.

Although the alkylated-subunits were fully active in cell free systems, a >99.5% reduction in alkylated-ricin cytotoxicity was determined in a cell culture, in which case the binding of the monomeric B subunit to the cell surface cannot promote internalization of the detached A subunit. This dramatic change in ricin-induced cytotoxicity, together with an in vivo test in mice, in which no body weight loss nor any noticeable side effect were observed, allowed the safe usage of the monomerized ricin for immunizing a horse.

Following immunization, high and stable ELISA and neutralizing antibody titers, were reached, and F(ab’)_2_-based anti-ricin antitoxin was produced from the hyperimmune horse plasma. It was previously claimed [5], that fractionation of anti-ricin IgG antibodies to F(ab’)_2_ could severely affect the neutralizing capabilities of the latter in vivo. Preliminary experiments conducted by us (data not shown) have demonstrated that the F(ab’)_2_ fragment possess the same potency as its IgG precursor both in vitro and in vivo. Indeed, the F(ab’)_2_-based antitoxin was found highly potent. The ED_50_ of the antitoxin in cell culture (~0.7 nM) was comparable to the ED_50_ of rabbit derived polyclonal anti-ricin IgG fraction [4]. These neutralizing capabilities at the nanomolar range are a property of highly efficient antibodies. In line with this, the antitoxin was found highly protective in mice, following administration at 24 h post intranasal intoxication with a lethal challenge of ricin. The survival ratios were >60% (whether the antitoxin was administered intranasally or intravenously), much higher than those obtained with the previously characterized rabbit-derived antibodies (~35%) [20,22]. Furthermore, high survival rates (34%) were obtained at a very late treatment time point, namely 48 h following intoxication. In sharp contrast to this, negligible surviving ratios (4%) were obtained when mice were treated with rabbit-derived anti-ricin antibody at 48 h post exposure [21]. In fact, the survival percentages of intoxicated mice treated at 24 h post exposure (PE) with horse-derived antitoxin were similar to those obtained following a combinational treatment with rabbit-derived anti-ricin antibody and immunomodulatory drugs [20,22], while the survival ratios following treatment at 48 h with horse antitoxin, were comparable to those obtained for mice treated with rabbit antibody at 24 h PE. This difference in protection is also reflected in the fact that treatment with the horse-derived F(ab’)_2_-based antitoxin induces a significant reduction in inflammatory parameters in comparison to rabbit-derived antibody-based treatment. Thus, while administration of horse antitoxin in itself induced a sharp attenuation in IL-6 levels (~90% reduction) at 72 h PE, rabbit antibody treatment had virtually no effect on IL-6 levels [25]. An additional outstanding effect of the horse antitoxin-based treatment shown in the present study was a significant reduction (by ~60%) of XO levels in the BALF of ricin-intoxicated mice. XO is an important source of reactive oxygen species (ROS) formation [26,27], which contributes significantly to pulmonary edema formation in diverse lung pathologies [28,29]. In contrast, we found that rabbit antibody-based treatment was not effective in reducing the levels of XO (data not shown). Edema markers were also dramatically reduced following treatment with the horse antitoxin (60% reduction in BALF protein content and 75% reduction in ChE levels at 72 h PE), in contrast to insignificant changes following treatment with rabbit antibody [25]. Indeed, when we previously abolished neutrophil-induced lung injury via total body irradiation (TBI), treatment at 48 h with rabbit antibody resulted in 42% survival rates, in parallel to sharp attenuation of IL-6 and edema marker levels [21].

In summary, in the present study we demonstrate that neutralized ricin antigen obtained by toxin monomerization and alkylation is completely safe for immunization, and induces high titers of potent ricin neutralizing antibodies that can be effectively used for passive immunization. F(ab’)_2_-based ricin antitoxin produced from the hyperimmune plasma of a horse that was immunized with monomerized ricin, conferred high level protection following pulmonary intoxication, which is the most fatal exposure route for ricin intoxication. This horse antitoxin-based treatment displayed effective anti-edematous and anti-inflammatory activities, apparently via balancing cytokine levels toward inflammatory attenuation in the lungs of intoxicated animals. We note that residual damage marker levels in the F(ab’)_2_-based ricin antitoxin-treated mice, nevertheless remained higher at 72 h following intoxication in comparison to the levels measured in naïve mice, suggesting that survival ratios may be further improved by combinational treatment with immunomodulatory drugs.

## 4. Materials and Methods

### 4.1. Ricin Preparation

Crude ricin was prepared from seeds of endemic *R. communis*, essentially as described before [30]. Briefly, seeds were homogenized in a Waring blender in 5% acetic acid/ PBS. The homogenate was centrifuged and the clarified supernatant containing the toxin was subjected to ammonium sulfate precipitation (60% saturation). The precipitate was dissolved in PBS and dialyzed extensively against the same buffer. The toxin preparation appeared on a Coomassie Blue stained non-reducing 10% polyacrylamide gel as 2 major bands of molecular weight approximately 65 kDa (=ricin toxin, ~80%) and 120 kDa (=ricinus communis agglutinin (RCA), ~20%). Pure toxin was prepared as described previously [30,31]. Briefly, under laminar flow and aseptic conditions the crude ricin preparation was loaded onto a gel-filtration column (Superdex 200HR 16/60 Hiload prep grade on an AKTA explorer, GE Healthcare Bio-Science AB, Uppsala, Sweden) and washed out with PBS to yield two protein peaks, corresponding to RCA and ricin. The purity of the ricin fraction was estimated by SDS-PAGE analysis to be >98%. Protein concentration was determined by 280 nm absorption (Nanodrop). 

### 4.2. Reduction and Alkylation of Pure Ricin

Monomerized ricin vaccine was prepared, essentially as previously described [13]. Briefly, pure ricin was incubated with 50 mM Dithiothreitol (DTT, Sigma-Aldrich, Rehovot, Israel) for 2 h in room temperature, the ricin-DTT solution was incubated for additional 2 h (room temperature, protected from light) with 100 mM Iodoacetamide (IAA, Sigma-Aldrich, Rehovot, Israel), and the product (alkylated-ricin) was extensively dialyzed against PBS. 

### 4.3. Gel Electrophoresis

Samples were visualized using Coomassie Blue stained non-reducing 10% polyacrylamide gel that was subjected to sodium dodecylsulphatepolyacrylamide gel electrophoresis (SDS-PAGE) under non-reducing conditions.

### 4.4. UPLC

Five µL samples (alkylated-ricin, alkylated-RTA or alkylated-RTB) were analyzed with a Water Acquity UPLC (Waters Corporation, Milford, MA, USA) equipped with a UV detector and binary solvent manager. The output signal was monitored and processed using Empower software (Empower 2.0, Waters Corporation, Milford, MA, USA). The method was employed using an Acquity UPLC BEH C-4 1.7 µm (2.1 × 50 mm) column (Waters Corporation, Milford, MA, USA). The flow rate of the mobile phase was 0.4 mL/min. The column temperature was 50 °C, and the eluted products were monitored at a wavelength of 215 nm. The samples were rinsed for 4.5 min in an acetonitrile gradient from 70% buffer A (5% acetonitrile in 0.1% trifluoroacetic acid [TFA]) and 30% buffer B (80% acetonitrile in 0.1% TFA), to 30% buffer A.

### 4.5. Isolation and Purification of Alkylated-Ricin Subunits

Monomerized ricin preparation was loaded on an α-Lactose-Agarose (Sigma-Aldrich, Rehovot, Israel) column, and extensively washed with PBS for collection of alkylated-RTA. Alkylated-RTB was eluted from the column with 0.5 M galactose. The purity of the isolated subunits was verified by UPLC.

### 4.6. Assessment of Ricin Activity in a Cell-Free Translational Assay

The biological activity of ricin was determined in a cell-free assay, as previously described [14,15]. Briefly, rabbit reticulocyte lysate containing luciferase mRNA was used to measure the activity of ricin via inhibition of protein synthesis. The relative biological activity was determined by comparing the luminescence levels in treated samples versus those of untreated controls. The amount of luciferase translated is inversely proportional to the activity of ricin.

### 4.7. Assessment of Ricin Activity in a Cell Culture

Genetically engineered HEK-293-acetylcholinesterase (AChE) cells [16] were cultured in Dulbecco’s modified Eagle’s medium (DMEM) (Biological Industries, Beit Haemek, Israel) supplemented with 10% fetal bovine serum (FBS). For the cytotoxicity studies, the cells were seeded in 96-well plates (0.75 × 10^5^ cells/well) in medium containing different concentrations of intact or monomerized ricin. Sixteen hours later, the medium was replaced, the cells were incubated for 2 h, and the amount of secreted AChE in each well was assayed according to Ellman et al. [32] in the presence of 0.1 mg/mL bovine serum albumin (BSA), 0.3 mM 5,5’-dithiobis(2-nitrobenzoic acid), 50 mM sodium phosphate buffer (pH 8.0), and 0.5 mM acetylthiocholine iodide (ATC) (Sigma-Aldrich, Rehovot, Israel). The assay was carried out at 27 °C and monitored by a Thermomax microplate reader (Molecular Devices, Ramsey, MN, USA).

### 4.8. In Vitro Ricin Neutralization Assay

HEK-293-AChE cells [16] were seeded in 96-well plates (0.75 × 10^5^ cells/well) in medium containing 2 ng/mL ricin, in the absence or presence of increasing doses of the F(ab’)_2_-based anti-ricin antitoxin. Sixteen hours later, the medium was replaced, the cells were incubated for 2 h, and the amount of secreted AChE in each well was assayed as described above.

### 4.9. Animal Studies

Animal experiments were performed in accordance with the Israeli law and were approved by the Ethics Committee for animal experiments at the Israel Institute for Biological Research (Horse protocol identification code: H-01-2015, date of approval: 9 August 2015; Rabbits protocol identification codes: RB-20-2011, RB-36-2012, dates of approval: 30 August 2011, 20 November 2012, respectively; Mice protocol identification code: M-59-2016, date of approval: 8 September 2016). Treatment of animals was in accordance with regulations outlined in the USDA Animal Welfare Act and the conditions specified in the Guide for Care and Use of Laboratory Animals (National Institute of Health, 1996). Local horse was immunized to produce the antitoxin preparation. New Zealand white rabbits (Charles River Laboratories Ltd., Canterbury, UK) weighing 2.5 to 3 kg were immunized in order to produce rabbit-derived polyclonal anti-ricin antibodies. Female CD-1 mice (Charles River Laboratories Ltd., UK) weighing 27–32 g were used for toxicity, survival and pathogenesis studies.

Prior to all studies in mice and rabbits, the animals were habituated to the experimental animal unit for 5 days. All mice were housed in filter-top cages in an environmentally controlled room and maintained at 21 ± 2 °C and 55 ± 10% humidity. Lighting was set to mimic a 12/12 h dawn to dusk cycle. Animals had access to food and water ad libitum.

### 4.10. Rabbit Anti-Ricin Hyperimmune Serum Production

Rabbits were immunized with native- or reduced- ricin in a stepwise manner, injections 1, 2, 3, and 4 containing 0.5, 4, 16, and 100 µg toxin/rabbit, with 4-week intervals between injections. Blood samples were collected (1 week after injection) to ascertain anti-ricin antibody titer build-up. Immunization was continued over 16 weeks, until steady high anti-ricin titers were observed.

### 4.11. Safety Studies in Mice

Mice were injected intraperitoneally with 40 µg/kg monomerized ricin, at a volume of 200 µL. Mice body weights were monitored for 14 days. In addition, careful individual detailed clinical examinations were carried out. The observations included, among others, changes in skin, fur, eyes, and occurrence of secretions and excretions. Changes in posture and autonomic activity (lacrimation and piloerection) and the presence of bizarre behavior were also checked.

### 4.12. In Vivo Ricin Neutralization Determination

Mice were injected intramuscularly with different volumes of rabbit-derived anti-ricin antisera, and 24 h later, the mice were intramuscularly intoxicated with a lethal dose of ricin. Preceding these studies, we determined that 15 µg crude ricin/kg body weight is approximately equivalent to 1 mouse (intramuscular) LD_50_. Mortality was monitored over 14 days.

### 4.13. Survival Experiments in Mice

For intranasal intoxication, mice were anesthetized by an intraperitoneal (i.p.) injection of ketamine (1.9 mg/mouse, Vetoquinol, Lure, France) and xylazine (0.19 mg/mouse, Eurovet Animal Health, AD Bladel, The Netherlands). Crude ricin (50 µL; 7 µg/kg diluted in PBS) was applied intranasally (2 × 25 µL) and mortality was monitored over 14 days. Preceding these studies, we determined that 3.5 µg crude ricin/kg body weight is approximately equivalent to 1 mouse (intranasal) LD_50_. Treatments via the intranasal route were performed on mice anesthetized as above. For antibody treatment following intranasal exposure to ricin, anti-ricin antitoxin was delivered intranasally or intravenously at 24 or 48 h following intoxication.

### 4.14. Bronchoalveolar Lavage Fluid (BALF) Analysis

Mice BALF were collected by instillation of 1 mL PBS at room temperature and were centrifuged at 3000 rpm at 4 °C for 10 min. Supernatants were collected and stored at −20 °C until further use.

BALF levels of IL-6 were determined by ELISA (R&D systems, Minneapolis, MN, USA). Cholinesterase (ChE) enzymatic activity was measured according to Ellman et al. [32]. Assays were performed in the presence of 0.5 mM acetylthiocholine (Sigma-Aldrich, Rehovot, Israel), 50 mM sodium phosphate buffer pH 8.0 (Sigma-Aldrich, Rehovot, Israel), 0.1 mg/mL BSA (Sigma-Aldrich, Rehovot, Israel), and 0.3 mM 5,5′-dithiobis-(2-nitrobenzoic acid) (Sigma-Aldrich, Rehovot, Israel). The assay was carried out at 27 °C and monitored by a Thermomax microplate reader (Molecular Devices, Ramsey, MN, USA). Protein levels in BALF were determined by 280 nm absorption (NanoDrop 2000, ThermoFisher Scientific, Waltham, MA, USA). Xanthine oxidase (XO) in BALF was determined by an activity assay kit (Molecular Probes, Eugene, OR, USA).

### 4.15. Horse Vaccination and Plasmapheresis

A horse was immunized with escalating doses of the monomerized ricin until a minimal level of neutralizing antibodies titer is elicited. The first three doses were 2, 5 and 10 milligrams in Incomplete Freund’s adjuvant (Statens Serum Institute, Copenhagen, Denmark). The following doses were adjuvant-free, at doses of 10 mg. Plasmapheresis of the hyperimmunized horse was conducted every three months using veterinary plasmapheresis instrument (plasma collection system, PCS-2, Haemonetics Corporation, Braintree, MA, USA). Ten liters of plasma were collected during each plasmapheresis procedure and plasma bags were stored at −20 °C.

### 4.16. F(ab’)_2_-Based Antitoxin Production

Concentrated anti-ricin F(ab’)_2_ preparations were generated from pooled horse hyperimmune antisera by one hour of pepsin (1200 U/mL, pepsin A from porcine stomach mucosa, Sigma, Steinhaim, Germany) cleavage of the Fc fragments at 30 °C and pH 3.2. The cleavage step was finalized by increasing the solution pH to 7.4 and reducing the temperature to 18 °C. Purification of the F(ab’)_2_ fragments was carried out in several stages: First, the contaminating proteins were precipitated with ammonium sulfate (25% saturation for 20 h at 18 °C). Then, the sediment that contains contaminating proteins was separated from the suspension that contains the F(ab’)_2_ fragments using crossflow microfiltration cassettes 0.2 µm (Tangenx technology, Shrewsbury, MA, USA). The filtrate of the microfiltration was washed (from small contaminating proteins and peptides), concentrated, and dialyzed (against 50 mM phosphate buffer pH 8) with crossflow ultrafiltration 30KD cassettes (Sartocon Cassette polyethersulfon 30KD, Sartorius, Goettingen, Germany). The dialyzed solution was applied on a Q-sepharose anion-exchange column (Q sepharose fast flow, GE healthcare, Uppsala, Sweden). The F(ab’)_2_ fragments fraction eluted with the flow through was collected, while the contaminating proteins remained bound to the column, which were then regenerated with 1 M NaCl. The flow through solution was adjusted to pH 6 and applied onto a SP-sepharose cation-exchange column (SP sepharose fast flow, GE healthcare, Uppsala, Sweden). The F(ab’)_2_ fragments fraction was eluted with the flow through and were collected, while, the contaminating proteins remained bound to the column, which were then regenerated with 1 M NaCl. All the chromatographic processes were carried out using an AKTA Process Instrument (GE healthcare, Uppsala, Sweden). Finally, under grade A conditions, the antitoxin solution was concentrated and dialyzed (against 300 mM glycine buffer, pH 7.4), and filtered with nanofilter (Kleenpak-Nova, PALL life science, MI, USA).

### 4.17. Statistical Analysis

Individual groups were compared using unpaired *t* test analysis. To estimate p values, all statistical analyses were interpreted in a two-tailed manner. Values of *p* < 0.05 were considered to be statistically significant. Kaplan–Meier analysis was performed for survival curves. All data is presented as means ± SEM.

## Figures and Tables

**Figure 1 toxins-10-00466-f001:**
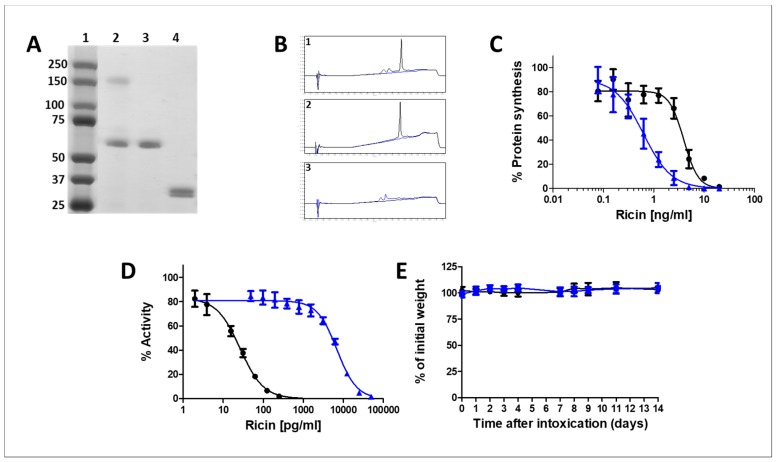
Characterization of the monomerized ricin vaccine. (**A**) SDS-PAGE analysis of the ricin purification and monomerization process (0.25 µg/lane) samples. Lane 1, Marker; Lane 2, Crude ricin; Lane 3, Purified ricin; Lane 4, Monomerized ricin. (**B**) UPLC chromatograms of alkylated-ricin and its purified subunits. 5 µL of the tested samples were injected into the UPLC columns (2.1 × 50 mm) and eluted at a flow rate of 0.4 mL/min. The chromatograms were monitored at 215 nm. 1. Alkylated-ricin preparation. 2. Alkylated-RTA. 3. Alkylated-RTB. (**C**) Catalytic activity assessment of ricin and monomerized ricin in a cell free system. The catalytic activities of purified ricin holotoxin (black line) and monomerized ricin (blue line) were determined using the transcription and translation (TnT) assay. Luminescence of untreated reticulocytes was considered as 100% protein synthesis. (**D**) In vitro activity of pure and alkylated ricin. Cultured HEK-293-AChE cells were incubated with increasing concentrations of pure (black line) and alkylated (blue line) ricin. The residual AChE activity in the culture medium was determined and expressed as the percent activity determined for untreated cells. (**E**) Toxicity following monomerized ricin administration to mice. Monomerized ricin (40 µg/kg body weight) was intraperitoneally administered to mice (*n* = 10), and body weights were determined at the indicated time points (blue line). Phosphate buffered saline (PBS) injected mice served as control (black line). Animals were observed for a 14 days period after alkylated ricin or PBS were injected.

**Figure 2 toxins-10-00466-f002:**
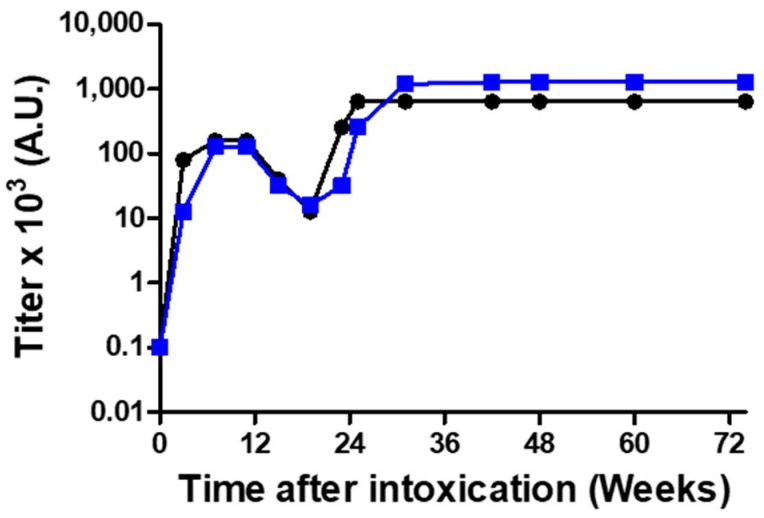
Titer buildup in the horse serum after vaccination with monomerized ricin immunization. The reactivity profile of the antibodies elicited by immunization were determined by enzyme linked immunosorbent assay (ELISA, black curve) and by in vitro ricin neutralization assay (blue curve).

**Figure 3 toxins-10-00466-f003:**
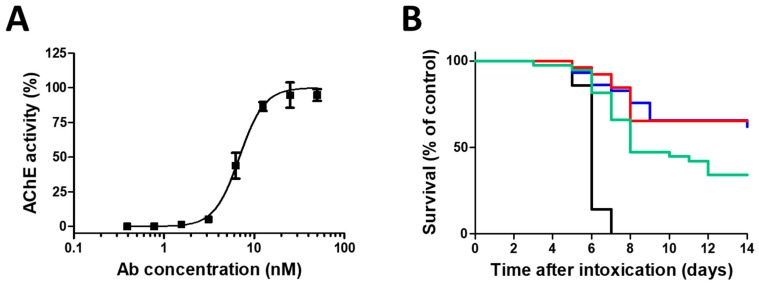
In vitro and in vivo efficacy of the anti-ricin F(ab’)_2_-based antitoxin. (**A**) In vitro ricin neutralization. Ricin (2 ng/mL) was mixed with increasing concentrations of the antitoxin. The mixtures were added to cultured HEK-293-AChE cells, and the residual AChE activity in the culture medium was determined 18 h later. (**B**) Kaplan–Meier survival curves of mice intoxicated with ricin and subjected to anti-ricin antibody treatment. Mice intranasally intoxicated with ricin (7 µg/kg body weight) were not treated (black line; *n* = 14), treated intranasally at 24 h post exposure (blue line; *n* = 29), treated intravenously at 24 h post exposure (red line; *n* = 26), or treated intravenously at 48 h post exposure (green line; *n* = 38) with horse derived F(ab’)_2_ anti-ricin antitoxin. Animals were observed for a 14-day period after ricin challenge.

**Table 1 toxins-10-00466-t001:** Anti-ricin antibody titers and protective doses of anti-ricin antisera elicited in immunized rabbits.

Antigen	ELISA-Antibody Titer ^a^	Neutralizing-Antibody Titer ^a^	PD_50_ (μL) ^b^
Native Toxin	256,000 ± 41,300	76,800 ± 14,300	1.1 ± 0.31
Monomerized Toxin	480,000 ± 71,500	96,000 ± 14,300	0.8 ± 0.14

^a^ Values present the average ± standard error (SEM) of 5–6 rabbits. ^b^ Mice were treated with the respective antisera 24 h prior to challenge with 2LD_50_ (30 µg/kg) ricin. Values present the average ± standard error (SEM) of 3 mice or 6 mice (treated with antisera harvested from rabbits immunized with native or monomerized toxin, respectively).

**Table 2 toxins-10-00466-t002:** Pro-inflammatory and damage markers in the bronchoalveolar lavage fluids (BALFs) of mice following ricin intoxication.

	Treatment Group		
	Naïve ^a^	Ricin ^b^	Ricin + Antitoxin ^c^
**IL-6 (pg/mL)**	0 ± 0	3547 ± 1372 **	451 ± 532 ^&&^
**Protein (mg/mL)**	0.5 ± 0.1	5.7 ± 1.8 **	2.1 ± 0.8 **^&&^
**ChE (mU/mL)**	0 ± 0	306 ± 93 **	68 ± 31 **^&&^
**XO (mU/mL)**	0.6 ± 0.1	4.1 ± 1.5 **	1.7 ± 0.7 *^&&^

* *p* < 0.05 between tested group and naïve; ** *p* < 0.01 between tested group and naïve; ^&&^
*p* < 0.01 in comparison to ricin intoxicated mice. ^a^
*n* = 4; ^b^
*n* = 6; ^c^
*n* = 5. BALF samples were collected from ^a^ naive mice, or ^b,c^ ricin intoxicated mice 72 h following exposure.

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
