# Peer review of "Generation of Highly Efficient Equine-Derived Antibodies for Post-Exposure Treatment of Ricin Intoxications by Vaccination with Monomerized Ricin"

_toxins, 2018, doi:10.3390/toxins10110466_

Round 1
Reviewer 1 Report
The authors demonstrate a new method of ricin inactivation for vaccine use. They have carefully characterized the vaccine, and the resulting immune response. Immunized animals make high titers of neutralizing and protective antibodies. They clearly show the anti-inflammatory effects of the antibodies, and good post-exposure protection.
I have two, relatively minor, editorial comments:
In lines 9-10 (Abstract) it is stated “The aim of this work was to generate a high affinity anti-ricin antitoxin.” But in fact, neither affinity nor even avidity was ever tested. Then in the Discussion (lines 253-64) they indicate that high-titered neutralization is “a property of high affinity antibodies”, suggesting that their antibodies must therefore be high affinity. The statement in the discussion is merely an assumption in this particular case. If you are not going to test affinity, then it is best to avoid making any claims of “high affinity”.
In lines 57-58 of the Introduction it is stated “generating a stable, non-toxic, monomerized ricin preparation for animal immunization.” As per figure 1D, it is not non-toxic, rather it has reduced toxicity.
Author Response
A reply to reviewer 1:
All changes, in accordance with your comments, are marked in green.
1. We thank the reviewer for pointing out the misleading use of the term affinity. Accordingly, we have changed the sentences in both the abstract (deleting the words "high affinity") and the discussion (changing "high affinity" to "highly efficient").
2. As for your comment regarding the toxicity of the monomerized ricin, we have now replaced in the Introduction section the term "non toxic monomerized ricin" with "generating a stable monomerized ricin preparation for animal immunization with substantially reduced toxicity".
Reviewer 2 Report
Manuscript ID: toxins-386652
Type of manuscript: Article
Title: Generation of highly efficient equine-derived antibodies for
post-exposure treatment of ricin intoxications by vaccination with
monomerized ricin
Submitted to section: Plant Toxins
Review:
This manuscript describes the development and testing of a novel ricin neutralizing antibody preparation for treatment of ricin toxicity. The authors create a ricin A-chain molecule that differs from the native form in being far less toxic to the animals from which neutralizing antibodies are to be derived, and to the technicians performing the work, but retaining sufficient structural similarity to elicit a strong antibody response, such that derived antibodies are protective against in vivo ricin challenge. The work is clearly explained and thoroughly investigated. All experiments are scientifically sound, and the results are convincing. This work represents important results for countering ricin toxicity.
Questions:
1) Lines 279-284: Was the outstanding effect of horse antitoxin-based treatment, as opposed to rabbit antibody-based treatment, on XO levels an expected result? Is this explained by the difference in species?
Small issues:
1) In Table 3, under “Ricin + antitoxic”, why does the measurement (451 +/- 532) not have a p value?
2) Line 238, “Plasmodium falciparum” should be in italics
3) Line 197, change “14-days” to “14-day”.
4) Line 439, change “on to” to “onto” (remove space).
5) I did not see where UPLC was written out in full when first introduced; occurs in Figure 1 and Methods 4.4.
6) Line 279, remove comma after “treatment”.
Author Response
A reply to reviewer 2:
All changes in accordance with your comments are marked in light blue.
As for the issue regarding the differences in xanthine oxidase levels, we note that XO levels were measured following either horse- or rabbit- derived antitoxin treatments in an identical manner. To our opinion, the difference most probably derives from the better neutralization activity of the horse antitoxin as opposed to rabbit antitoxin.
All other small issues raised by you were changed (UPLC was introduced within the text).
In Table 3, under “Ricin + antitoxic”, why does the measurement (451 ± 532) not have a p value?
Our response:
There is no significant difference (p=0.14) between the “Ricin + antitoxin” group and the "naïve" group regarding the IL-6 levels in Bronchoalveolar lavage fluids (451 ± 532 vs 0 ±0).
In contrast there is a significant difference (p<0.01) between “Ricin + antitoxin” group and "ricin" group.